# Replacing critical point drying with hexamethyldisilazane drying enhances the ultrastructural preservation of cell surface projections in the parasite *Trichomonas vaginalis* for scanning electron microscopy

**Tuanne dos Santos Melo[1], Abigail Miranda-Magalhães[1], Regina Celia Bressan Queiroz de Figueiredo[2]ᴑ\*, Antonio Pereira-Neves****[2]ᴑ\***

**1** Fiocruz, Programa de Pós-graduação em Biociências e Biotecnologia em Saúde, Instituto Aggeu Magalhães, Recife, Pernambuco, Brazil, **2** Departamento de Microbiologia, Fiocruz, Instituto Aggeu Magalhães, Recife, Pernambuco, Brazil

ᴑ These authors contributed equally to this work.
\* antonio.neves@fiocruz.br (AP-N); regina.bressan@fiocruz.br (RCBQF)

## Abstract

The cell surface of *Trichomonas vaginalis*, causative agent of human trichomoniasis, plays a pivotal role in parasite adhesion, motility, and intercellular communication. Scanning electron microscopy (SEM) is widely used to explore the surface projections involved in these processes; however, standard sample preparation via critical point drying (CPD) often damages delicate membrane projections. Here, we optimized a hexamethyldisilazane (HMDS) drying protocol as a reliable alternative to CPD for ultrastructural analysis of *T. vaginalis* using SEM. Both drying methods were compared in terms of image quality and artifact formation. HMDS drying significantly improved the preservation of fragile projections, such as filopodia-, cytoneme-, and lamellipodia-like structures, compared to CPD. Our results show that susceptibility to CPD-induced artifacts may vary among highly adherent *T. vaginalis* strains, highlighting the need for caution in SEM interpretation. In a strain previously CPD-characterized by exhibiting a low number of cytonemes, quantitative analyses revealed a marked increase in the number of parasites with filopodia and cytonemes upon HMDS, accompanied by a reduction in drying-associated artifacts. In contrast, other strains exhibited similar quantitative results for both methods, though HMDS demonstrated a slight qualitative enhancement. HMDS drying also enabled the identification of novel ultrastructural features of *T. vaginalis*, including (a) long (>20 µm) cytonemes forming network-like connections between parasites and host cells, and (b) thin posterior and axostylar cytoneme-like structures that seems involved in host cell adhesion. Moreover, HMDS provided better morphological preservation of amoeboid forms and enhanced visualization of parasite–host interactions, revealing membranous networks not previously observed with CPD. Altogether, this study

**Data availability statement:** All relevant data are within the manuscript and its Supporting information files.

**Funding:** This research was supported with grant from Fiocruz PROEP-IAM IAM-005-FIO-22-2-13 and Coordenação de Aperfeiçoamento de Pessoal de Nível Superior - Brasil (CAPES) - Finance Code 001. TSM and AMM are PhD fellow from Coordenação de Aperfeiçoamento de Pessoal de Nível Superior (CAPES). The funders had no role in study design, data collection and analysis, decision to publish, or preparation of the manuscript.

**Competing interests:** The authors have declared that no competing interests exist.

demonstrates the importance of the drying methods for sample preparation and expands the approaches for parasite imaging, revealing HMDS as a valuable option that could provide new ultrastructural insights into the surface morphology and intercellular communication mechanisms of *T. vaginalis.*

## Introduction

The protist *Trichomonas vaginalis* (Parabasalia) is an important yet often neglected pathogen responsible for trichomoniasis, the most widespread non-viral sexually transmitted infection in humans, with an estimated 156 million new cases worldwide in 2020 [1,2]. This accounts for about half of the global incidence of infections caused by all non-viral sexually transmitted parasites [1]. *T. vaginalis* infections, while commonly asymptomatic, can range from mild irritation to severe inflammation in several regions of the urogenital tract in both women and men [2]. Untreated *T. vaginalis* can lead to serious complications, such as pelvic inflammatory disease, adverse birth outcomes, and infertility [3,4]. Although rare, perinatal transmission can occur, resulting in vaginal and respiratory infections in newborns [5]. Additionally, *T. vaginalis* infections are associated with an increased risk of HIV acquisition and urogenital cancers [6,7]. Despite its significant burden and health consequences, many aspects of the cell biology and pathogenicity of *T. vaginalis* remain understudied or unknown, e.g., role of extracellular vesicles and surface protrusions in parasite-to-parasite and parasite-to-host cell interaction, strain variability, and interaction with host microbiome and immunity.

As an extracellular pathogen, the cell surface of *T. vaginalis* is crucial for its interaction with the host cells, intercellular communication, and survival [8,9]. Therefore, a comprehensive understanding of the structural organization of the *T. vaginalis* surface is fundamental to advancing knowledge of the parasite's biology. In this context, scanning electron microscopy (SEM) is a useful tool that has been widely used for nearly 50 years to explore the finest features of *T. vaginalis* cell surface, unravelling its interaction with other microorganisms and hosts cells [10–14]. Over recent years, using SEM and light microscopy, our lab and others have identified novel structures and mechanisms, such as the biogenesis and release of extracellular vesicles [15–17] and the discovery of an extra-axonemal structure in trichomonads flagella [16], that play crucial roles in the parasite's communication and its ability to adhere to host cells and to other parasites, key steps in establishing *T. vaginalis* infection.

Expanding on these findings, we and others have demonstrated that some highly adherent *T. vaginalis* strains exhibit a wide array of surface motile protrusions that are implicated in critical biological processes such as phagocytosis, cytoadherence, migration, inter-parasite communication, and cytokinesis. These include: (a) lamellipodia-like projections, broad actin-based sheet-like extensions typically found at the leading edge [11,18]; (b) pseudopod-like protrusions, actin-rich tubular extensions with variable diameters and lengths [11,18–20]; (c) filopodia-like projections, thin finger-like actin-based structures generally emerging from lamellipodia or

pseudopods, measuring 0.1–0.3 µm in diameter with variable lengths [8,9,18,19]; (d) cytoneme-like structures, considered specialized and thinner filopodia of unknown molecular composition, generally <150 nm in diameter and capable of forming long inter-parasite connections [9,18]; (e) nanotube-like protrusions, tubulin-containing extensions morphologically similar to mammalian tunnelling nanotubes, ranging from ~7–32 µm in length and 300–500 nm in diameter [19]; and (f) uropod-like projections, posteriorly localized structures hypothesized to function as anchoring points during the attachment process [14].

However, visualizing these structures with a conventional high-vacuum SEM is not a trivial task; as common for most biological specimens, it generally requires meticulous processing to accurately preserve ultrastructural details as close as possible to their natural state. That involves cell fixation, dehydration in an intermediate solvent, such as ethanol or acetone, drying and sputter coating. A crucial challenge in preparing biological specimens for high-vacuum SEM is water removal; its presence can disrupt the microscope vacuum and cause severe artifacts that distort cell morphology. In this sense, critical point drying (CPD) is generally the most commonly used method for sample drying [21,22]. For this, water is first replaced with an intermediate solvent, which is then substituted with liquid carbon dioxide ($CO_2$). The $CO_2$ is brought to its critical point (31.1°C and 73.8 bar), where it to transition directly from liquid to gas without changing density, thus preventing the harmful surface tension effects that could compromise the sample's structural integrity [21,22].

Despite the wide usage of the CPD drying method, it is highly invasive and can change the shape and structure of biological samples due to the rapid changes in temperature, pressure, and osmolarity involved in the procedure [22,23]. Such conditions create a potential physical hazard for small, fragile cell surface projections, e.g., filopodia, cytonemes and tunneling nanotubes, that can lead to substantial and irreversible collapse of these structures [24–27]. In addition, CPD requires substantial preparation time, expensive equipment with high maintenance cost and a liquid $CO_2$ supply, which may be impractical for smaller laboratories [28]. Therefore, an alternative drying technique may be preferable to better preserve the ultrastructure of thin cell protrusions such as those of the T. vaginalis cell surface.

Hexamethyldisilazane (HMDS) has been employed as an alternative chemical drying agent for preparing a wide range of biological samples for SEM, including cells and soft tissues, usually achieving results comparable to or even superior to those obtained through CPD [23,24,29–38]. Unlike CPD, HMDS is a simple, rapid, and cost-effective method that does not require specialized equipment or advanced technical expertise, aside from standard precautions when handling highly toxic and hazardous chemicals, such as working within a safety fume hood. Despite its common use, the mechanism underlying HMDS drying remains poorly understood. It is hypothesized that its low surface tension, combined with protein cross-linking properties, strengthens the samples [31,36,39], enabling fragile cellular protrusions and filamentous structures to resist collapse during the evaporation process [29–32,34,40].

In this context, HMDS has also been used as an effective drying method for preserving the ultrastructure of various free-living, ruminal or parasitic protists, such as unicellular algae [41,42], amoebae [43], ciliates [44], kinetoplastids [45–50] and diplomonads [51]. Using this technique, new insights have been gained into the mechanism of extracellular vesicle secretion and formation of membrane extensions in these microorganisms, such as uroid filaments, nanotubes and filopodia [42,43,45,50]. Recently, our lab and others demonstrated through SEM of CPD-dried T. vaginalis cells that some highly adherent strains exhibit a greater number of cytoneme-like projections compared to poorly- or non-adherent parasites [8,9]. However, some surface protrusions have been inconsistently observed in certain CPD-dried highly adherent strains, such as the FMV1 strain [18,52]. Although lamellipodia- and pseudopod-like structures have been observed in the FMV1 strain [52], thinner projections such as cytonemes have been infrequently visualized by SEM [18].

We hypothesized that the CPD process may compromise the preservation of these delicate structures. To test this, we aimed to identify optimized HMDS drying conditions that could achieve an ultrastructural integrity of T. vaginalis comparable to or exceeding that of CPD-processed samples. Although Malli and colleagues have previously assessed HMDS drying for T. vaginalis, they reported that their protocol did not adequately preserve the parasite's ultrastructure, provoking several ultrastructural artifacts and the loss of distinct morphological features such as the flagella [53].

The present work expands upon previous SEM studies of *T. vaginalis* and provides both technical and ultrastructural insights. Here, we established an alternative, simple and time-efficient HMDS drying protocol for preparing *T. vaginalis* for SEM. Our protocol achieved a better preservation of surface projections compared to CPD, and revealed novel features not previously described by SEM, such as long (>20 μm) branched cytonemes forming networks between parasites and host cells, and posterior cytonemes potentially involved in adhesion. Our findings also suggest that the protrusions of a parasite strain can be fragile to CPD, highlighting the importance of employing multiple drying techniques to accurately characterize the surface morphology of different *T. vaginalis* isolates with minimal distortion or artifacts. Furthermore, our HMDS protocol is effective in preparing and visualizing the interactions of *T. vaginalis* with fibronectin and host cells, revealing membranous networks not previously observed with CPD and offering an alternative method for studying these cellular processes.

## Results and discussion

### Optimization of HMDS drying conditions for *T. vaginalis* ultrastructural preservation

HMDS-based protocols are typically established empirically and vary depending on sample characteristics, including size, composition, and the desired level of ultrastructural detail [54]. One of the most controllable parameters is the number of transitional steps between absolute ethanol and 100% HMDS [54]. To our knowledge, the study by Malli et al. [53] is the only published work comparing CPD and HMDS for *T. vaginalis*. Their protocol involved immersion in two successive HMDS baths for 10 minutes each, followed by 3 hours of evaporation at room temperature. However, they reported several ultrastructural artifacts and morphological alterations using this method [53].

Here, we initially tested three HMDS-based protocols with different transitional steps using axenic cultures of the FMV1 strain (S1–S3 Figs). In agreement with Malli et al. [53], all three protocols resulted in significant artifacts, including wrinkling or flattening surface, and the loss of distinct morphological features such as the anterior flagella, undulating membrane, and axostyle. Most of the parasites were coated with a thick HMDS film, often merging individual structures into a single continuous mass (S1–S3 Figs). Given that HMDS contains nonpolar methyl groups capable of film formation through wet deposition [55,56], the immersion of samples in large volumes (500 μL) of HMDS for extended periods (one or three baths for 10 min each), followed by slow evaporation, likely contributed to the observed artifact formation.

To address this, we modified Protocol 3 (S3 Fig) to reduce the sample's HMDS exposure time and promote more rapid solvent evaporation. In this optimized protocol (Protocol 4), following the final ethanol/HMDS transitional step, samples were briefly immersed in 100% HMDS (300 μL) for up to 30 seconds, gently blotted on filter paper to remove excess reagent, and air-dried for 30 minutes in a Petri dish (Fig 1A and S1 Movie). Based on the comparative results obtained across the tested protocols, the modified HMDS drying method (Protocol 4) represented the optimal balance between preparation time and preservation of structural integrity (Fig 1). This revised approach provided superior preservation of *T. vaginalis* surface morphology compared to CPD, as detailed in the subsequent sections.

The incubation time in the final HMDS bath revealed to be a critical factor for achieving optimal preservation of *T. vaginalis* morphology. In our assays, when samples were immersed in 100% HMDS for longer than 30 seconds, artifacts consistent with HMDS film deposition were observed similar to those shown in S1–S3 Figs. This finding is particularly unexpected, as most optimized HMDS-based protocols used for other protists, such as *Naegleria fowleri* [43], *Entamoeba gingivalis* [57], *Giardia* [51,58], *Trypanosoma brucei* [45,47,59], and *Leishmania* [60–62], employ final HMDS incubation times of at least 5 minutes. All these protocols are largely empirical, and at present, any mechanistic explanation for the need for such a brief final HMDS exposure in *T. vaginalis* would be purely speculative.

### HMDS drying provides a better preservation of the surface projections of *T. vaginalis* grown in axenic culture

Using the conventional CPD-based protocol, axenically cultured *T. vaginalis* FMV1 strain exhibited typical morphology, as previously described [52], including highly polymorphic forms—piriform, ellipsoid, elongated, spherical, and amoeboid cell

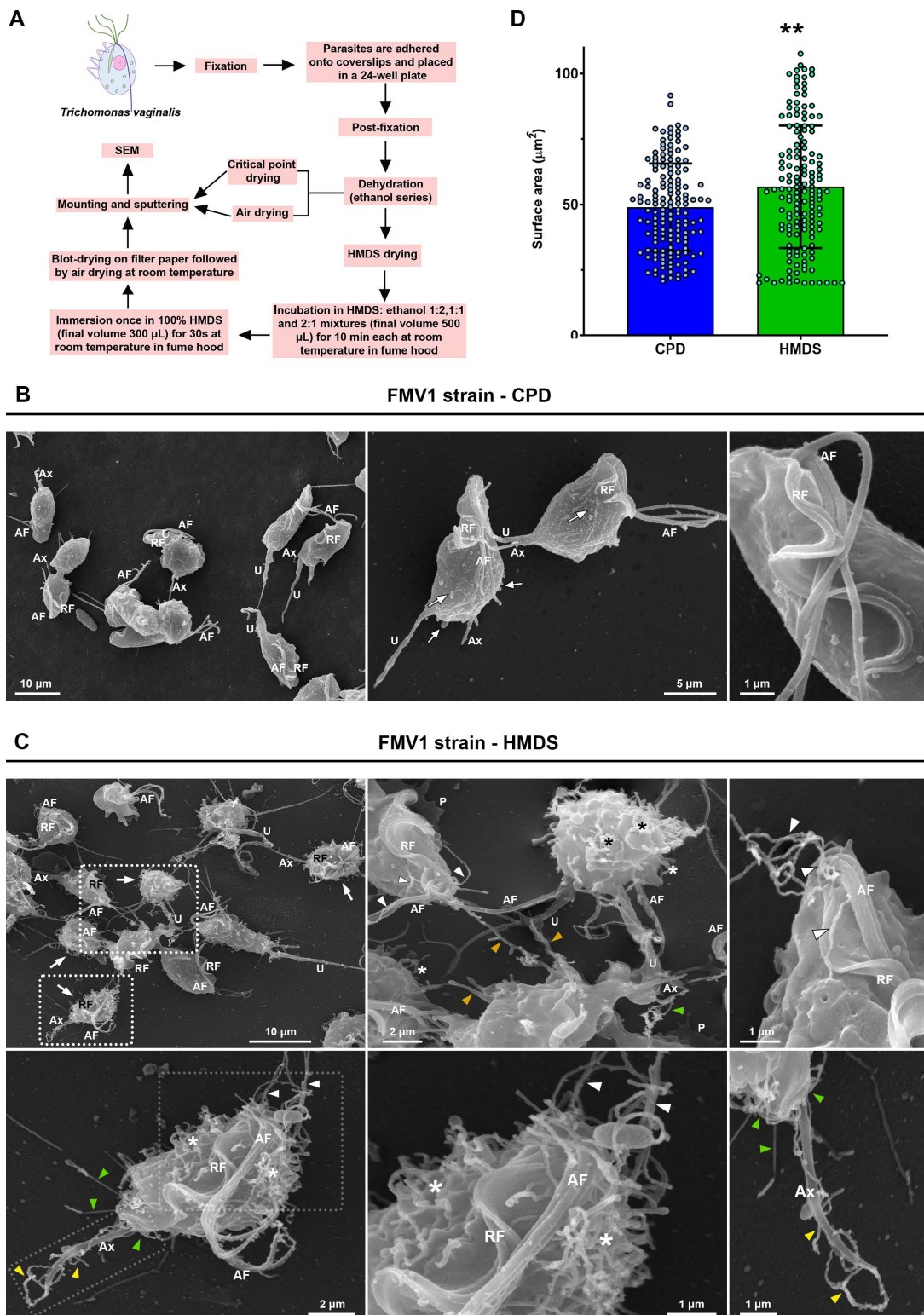

**Fig 1. Comparison of *T. vaginalis* (FMV1 strain) following CPD and HMDS drying procedures for SEM.** (A) Schematic representation of sample preparation procedures for SEM. (B–C) Representative SEM images illustrating morphological differences between parasites processed by CPD (B) and HMDS drying (C). In CPD-prepared samples (B), uropod-like structures (U) are observed protruding from the posterior pole, while microvesicle-like

structures and few small tubular projections (white arrows) are visible on the parasites' cell body; no projections are observed in the region where the anterior (AF) and recurrent (RF) flagella emerge. In contrast, HMDS-prepared samples (C) display a greater abundance of surface projections, including uropods (U) and numerous protrusions extending from the entire cell surface (white arrows). Cytoneme-like structures are observed emerging from the cell body (*), flagellar base region (white arrowheads), posterior pole (green arrowheads), and axostyle (Ax; yellow arrowheads). Filopodia (orange arrowheads) and pseudopods (P) are also seen. (D) Quantification of surface area in CPD- and HMDS-dried parasites. Bars represent the mean ± standard deviation from three independent experiments. SEM images of 50 randomly selected parasites per sample were analyzed using ImageJ software (area parameter). Individual points represent the measured surface area of each parasite. HMDS-dried parasites exhibit a significantly larger surface area compared to CPD-dried parasites. **$p < 0.01$, Mann–Whitney U test.

bodies—with slightly irregular surfaces displaying undulations and microvesicle-like structures (Fig 1B). Moreover, four anterior flagella, a recurrent flagellum associated with the undulating membrane, and an axostylar projection emerging from the posterior were clearly observed (Fig 1B). Uropod-like projections, a structure recently characterized at the posterior pole of adherent strains by our group [14], were also identified in some parasites (Fig 1B). In addition, few bulbous or tubular protrusions, ranging from 0.2 μm to 1.5 μm in diameter, were seen on the parasites' cell body; however, no projections were observed in the region where the anterior and recurrent flagella emerge (Fig 1B).

In contrast, using the optimized HMDS-drying protocol (Fig 1A, S1 Movie) not only preserved typical ultrastructural features such as the flagella, undulating membrane, and axostyle, but also revealed a myriad of surface projections—including uropods, pseudopods, filopodia, and cytonemes— extending from the entire parasite surface (Fig 1C). These tubular structures were found emerging from the cell body, posterior pole, axostyle, and the region surrounding the flagellar base (Fig 1C). It is important to note that in *T. vaginalis*, the term "filopodia" is broadly used to describe various types of tubular extensions protruding from the cell surface, with cytonemes regarded as thin, specialized filopodia [9,14]. In the absence of molecular markers to distinguish these structures, classification has been based solely on morphological criteria. We used the following: tubular projections with diameter < 150 nm were considered as cytonemes, while those > 150 nm were designated as filopodia.

The flagellar cytonemes displayed homogeneous morphology with diameters ranging from 70–100 nm (Fig 1C), similar to those previously reported in CPD-dried samples of CDC1132 strain of *T. vaginalis* [9], as well as the related veterinary trichomonad *Tritrichomonas foetus* [63]. In contrast, projections extending from the cell body exhibited greater heterogeneity, with diameters ranging from 70–800 nm (Fig 1C). Additionally, we occasionally observed bundles of thin cytoneme-like structures (<150 nm in diameter) with bulbous tips protruding from the posterior pole and axostylar region of FMV1 parasites (Fig 1C). To our knowledge, such structures have not been previously reported in *T. vaginalis* and closely resemble the uroid filaments described at the posterior pole of the amoeba *Naegleria fowleri* [43]. In contrast, parasites directly air-dried after ethanol dehydration displayed markedly morphological damage, including cell shrinkage, flagellar disruption and surface collapse (S4 Fig), highlighting HMDS's role in artifact reduction during the final air-drying stage of the protocol (Fig 1A and S1 Movie).

In mammalian cells, HMDS has been shown to better preserve the actin cortex compared to CPD, reducing cell shrinkage and preserving the cell structures and surface area as closely to the native state as possible [23,40]. To further compare the effects of CPD and HMDS on the ultrastructure of *T. vaginalis*, we quantified the surface area of FMV1 parasites submitted to both drying methods (Fig 1D). HMDS-dried parasites showed significantly larger mean surface area (56.7 ± 23.4 μm²) compared to those dried using CPD (48.9 ± 16.7 μm²) (Fig 1D). This difference likely reflects the greater abundance of surface projections in HMDS-treated samples, which contribute to the increased surface area. Furthermore, cell shrinkage and structural ruptures were observed in CPD-treated samples (S1 Fig), while these artifacts were markedly reduced in HMDS-treated parasites. Notably, projections such as pseudopodia were well preserved and remained firmly attached to the substrate in HMDS-dried cells (Fig 1C).

To rule out the possibility that the multiple filopodia- and cytoneme-like projections in HMDS-dried FMV1 parasites were artifacts induced by the compound, we compared both drying methods in two strains previously characterized by our

group via CPD: CDC1132, a highly adherent strain exhibiting numerous surface projections, and G3, a poorly adherent strain with few or no such protrusions [9,14]. Specifically, CDC1132 parasites displayed uropods, pseudopods, and multiple filopodia and cytonemes protruding from the flagellar base and cell body (Fig 2A, B), while no surface projections were found in G3 parasites upon both drying conditions (Fig 2C).

Nonetheless, HMDS treatment yielded slightly better quality of surface preservation in both strains compared to CPD. In CDC1132 parasites, almost no posterior and axostylar cytoneme-like protrusions were observed in CPD samples; only a few short tubular structures were found in the posterior pole of some parasites. In contrast, HMDS-dried CDC1132 parasites exhibited numerous and elongated posterior and axostylar cytonemes (Fig 2A), which may explain why such structures were not previously described in this strain using CPD-based preparation [9]. Moreover, projections such as pseudopods, uropods, filopodia and cytonemes in contact with adjacent parasites were occasionally found ruptured in CPD-dried CDC1132 strain, whereas such projections were intact in HMDS-dried parasites, even showing contact points between adjacent parasites' cytonemes (Fig 2B). In the G3 strain, parasites with a wrinkled surface were occasionally observed in CPD-prepared samples, whereas all HMDS-treated parasites displayed the expected slightly irregular surface with small undulations (Fig 2C).

Interestingly, SEM analysis suggests that the abundance of surface protrusions in HMDS-dried FMV1 parasites (Fig 1C) seems to be similar to that observed in both CPD- and HMDS-dried CDC1132 parasites (Fig 2A). To confirm this, we quantified the percentage of parasites displaying filopodia and/or cytonemes in the FMV1, CDC1132, and G3 strains upon CPD and HMDS drying (Fig 2D). In FMV1, 38% of HMDS-dried parasites exhibited such projections, in contrast to only 1% in CPD-treated samples. For the CDC1132 strain, similar percentages were recorded for both drying methods, with 51% and 55% of parasites exhibiting filopodia and/or cytonemes after CPD and HMDS procedures, respectively. In the G3 strain, which is poorly adherent, only 0.7% of parasites exhibited these projections in both drying methods. We highlight that, in each experimental replicate, samples from all three strains were placed in the same specimen holder and processed simultaneously within the CPD chamber, ensuring consistent experimental conditions for all samples.

Because both CPD and HMDS samples were derived from the same culture tube and submitted to identical conditions for fixation, post-fixation, dehydration, and sample handling, our data clearly indicates that filopodia/cytonemes of the FMV1 strain are more susceptible to damage associated with CPD compared to those in the CDC1132 strain. These findings suggest that, in addition to the strain-specific phenotype, the abundance of membrane protrusions observed on the surface of *T. vaginalis* by SEM is influenced, at least in part, by the drying method employed. This raises the question of why filopodia/cytonemes from different highly adherent strains exhibit variable sensitivity to CPD. One plausible explanation may lie in structural differences among these protrusions in distinct strains. Since most *T. vaginalis* surface projections are actin-rich structures [8, 11, 18–20], and actin expression has been shown to vary even among highly protrusive adherent strains [20], it is reasonable to hypothesize that differences in actin levels or organization within these structures may affect their stability under physical stress. Alternatively, variations in the biophysical properties of the parasite surface from distinct strains could contribute to the differential sensitivity of filopodia and cytonemes to CPD. For example, van der Waals forces might promote aggregation of these projections onto the cell surface, leading to structural deformation under high-pressure conditions, such as the worm-like structures previously reported in osteoblastic cells [27]. Similarly, we observed worm-like structures in CPD-dried FMV1 parasites, which might represent distorted or residual filopodia/cytonemes (S5 Fig). However, additional studies are necessary to investigate these hypotheses.

Although a significant quantitative difference in surface projections was observed in the FMV1 strain, a qualitative improvement in ultrastructural preservation was also evident in the other two strains following HMDS drying. Notably, CDC1132 parasites exhibited longer posterior and axostylar cytonemes, and no wrinkled cells were observed in the G3 strain (Fig 2A–C). These observations are consistent with previous reports in insect and mammalian cells, where delicate membrane protrusions—such as microvilli, filopodia, and cytonemes—are known to be highly susceptible to damage from physical stress during CPD or over-fixation [25–27,64]. Consequently, HMDS has emerged as a preferable alternative

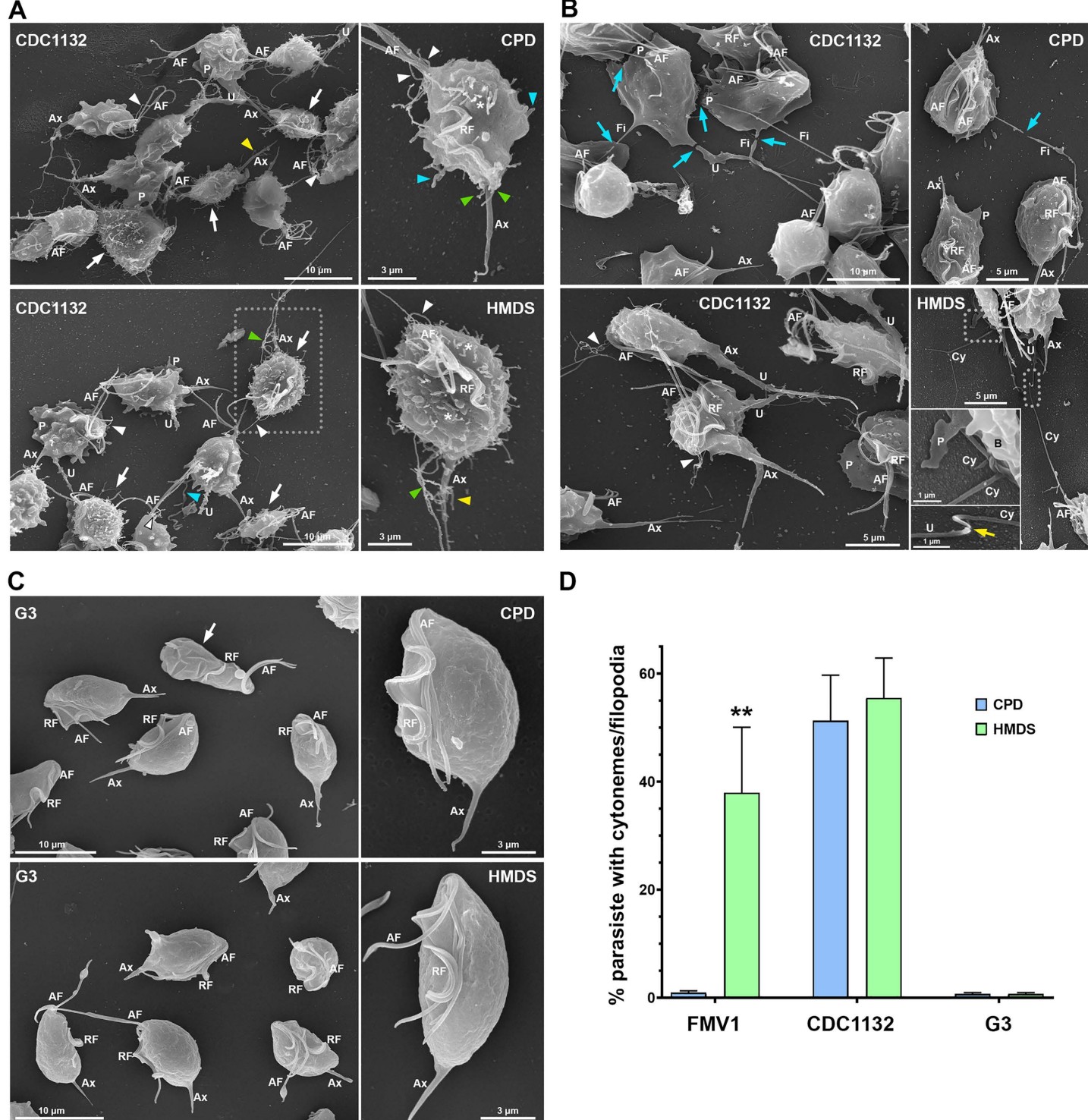

**Fig 2. Comparison of *T. vaginalis* (CDC1132 and G3 strains) following CPD and HMDS drying procedures for SEM.** (A, B) Representative SEM images of general and detailed views of the *T. vaginalis* CDC1132 strain processed by CPD and HMDS drying. (A) A myriad of surface projections is seen in parasites processed by both CPD or HMDS (white arrows). Cytonemes protruding from cell body and flagellar base region are indicated by asterisks and white arrowheads, respectively. Cytoneme-like projections emerging from the posterior pole and axostyle (Ax) are indicated by green and yellow arrowheads, respectively. In detail, the CPD-dried parasite displays only a few short tubular structures at the posterior pole, whereas the

HMDS-dried parasite exhibits longer and more abundant posterior and axostylar cytonemes. Pseudopodia (P) and filopodia (blue arrowheads) are also seen. (B) In CPD-dried sample, projections such as pseudopods (P), uropods (U) and filopodia (Fi) in contact with adjacent parasites are seen ruptured (blue arrows). In contrast, these structures remain intact in HMDS-dried samples. Insets show well-preserved cytonemes in contact with the parasite body (B) and a region of contact between the tips of two cytonemes from adjacent parasites (yellow arrow). Flagellar cytonemes are indicated by white arrowheads. (C) Representative SEM images of general and detailed views of the *T. vaginalis* G3 strain processed by CPD and HMDS drying. In both conditions, parasites display well-preserved, typical piriform or ellipsoid morphology with a slightly irregular surface; no protrusions are observed. A parasite with a wrinkled surface in the CPD-dried sample is indicated by a white arrow. (D) Quantification of the percentage of parasites from FMV1, CDC1132, and G3 strains exhibiting filopodia and/or cytonemes on their cell surface following CPD and HMDS drying. Bars represent the mean ± standard deviation from three independent experiments performed in duplicate. 500 parasites were randomly counted per sample using SEM. **$p < 0.01$, Mann–Whitney U test. AF, anterior flagella; RF, recurrent flagellum.

for preserving these fine structures, allowing for more accurate assessments of their length, abundance, and morphology [32,34,35,40].

Likewise, HMDS-based drying has proven effective in preserving cell projections involved in intercellular interactions in other protists, including uroid filaments in *Naegleria fowleri* [43], tubular extensions in marine microalgae such as *Dunaliella tertiolecta* and *Phaeodactylum tricornutum* [42], and nanotubes in *Trypanosoma brucei* [45]. Altogether, our results demonstrate that HMDS can be used as an alternative to CPD for preparing *T. vaginalis* grown in axenic culture for SEM, providing a better preservation of surface features, particularly filopodia- and cytoneme-like projections. Additionally, based on our data, we strongly recommend the combined use of CPD and HMDS drying procedures whenever feasible, to ensure a more comprehensive and artifact-minimized characterization of *T. vaginalis* surface structures by conventional SEM.

### Morphological aspects of surface projections of HMDS-dried FMV1 parasites during intercellular interactions in axenic culture

Recently, nanotubes, filopodia, and cytonemes have frequently been observed connecting adjacent *T. vaginalis* cells in axenic culture, suggesting a potential role for these surface projections in parasite-to-parasite communication [9,19]. As previously reported [65,66], the formation of clumps in *T. vaginalis* cultures generally correlates with a strain's ability to adhere to and exert cytotoxic effects on host cells. Specifically, highly adherent strains tend to aggregate in suspension when cultured in the absence of host cells [65,67], and cytonemes are commonly detected connecting individual parasites within these clumps [9].

Here, we took advantage of our HMDS-based drying protocol to visualize and characterize the morphological features of surface projections in the FMV1 strain during parasite-to-parasite interactions in axenic culture, including within clumps (Fig 3). Consistent with previous findings shown in the CDC1132 strain [9], our SEM analyses revealed that cytonemes and filopodia emerging from the anterior flagella and cell body were frequently observed connecting adjacent cells within clumps of *T. vaginalis* FMV1 parasites grown in suspension (Fig 3A). The distal ends of the cytonemes were frequently found in contact with various regions of adjacent parasites, including the cell body, axostyle, recurrent flagellum, or other cytonemes and filopodia (Fig 3A, B). To our knowledge, cytonemes contacting the recurrent flagellum have not been previously described in *T. vaginalis.* Interestingly, bloodstream forms of *Trypanosoma brucei* also produce membranous protrusions originating from the flagellar membrane that establish stable intercellular connections with the posterior end of other trypanosomes over long distances (>20 µm) [68].

Using the HMDS-based protocol, we similarly identified long cytonemes in *T. vaginalis*—extending up to 30 µm—that projected toward and contacted with other parasites (Fig 3C). These very long projections have not been previously reported in *T. vaginalis*. Quantitative analysis of 150 cytonemes revealed a highly variable length distribution, ranging from 0.2 µm to 30 µm, with a mean length of 3.62 ± 4.66 µm (S1 Dataset). Importantly, filopodia and cytonemes have not been previously described within parasite clumps of CPD-dried FMV1 cells grown in suspension [18,52]. In agreement with this,

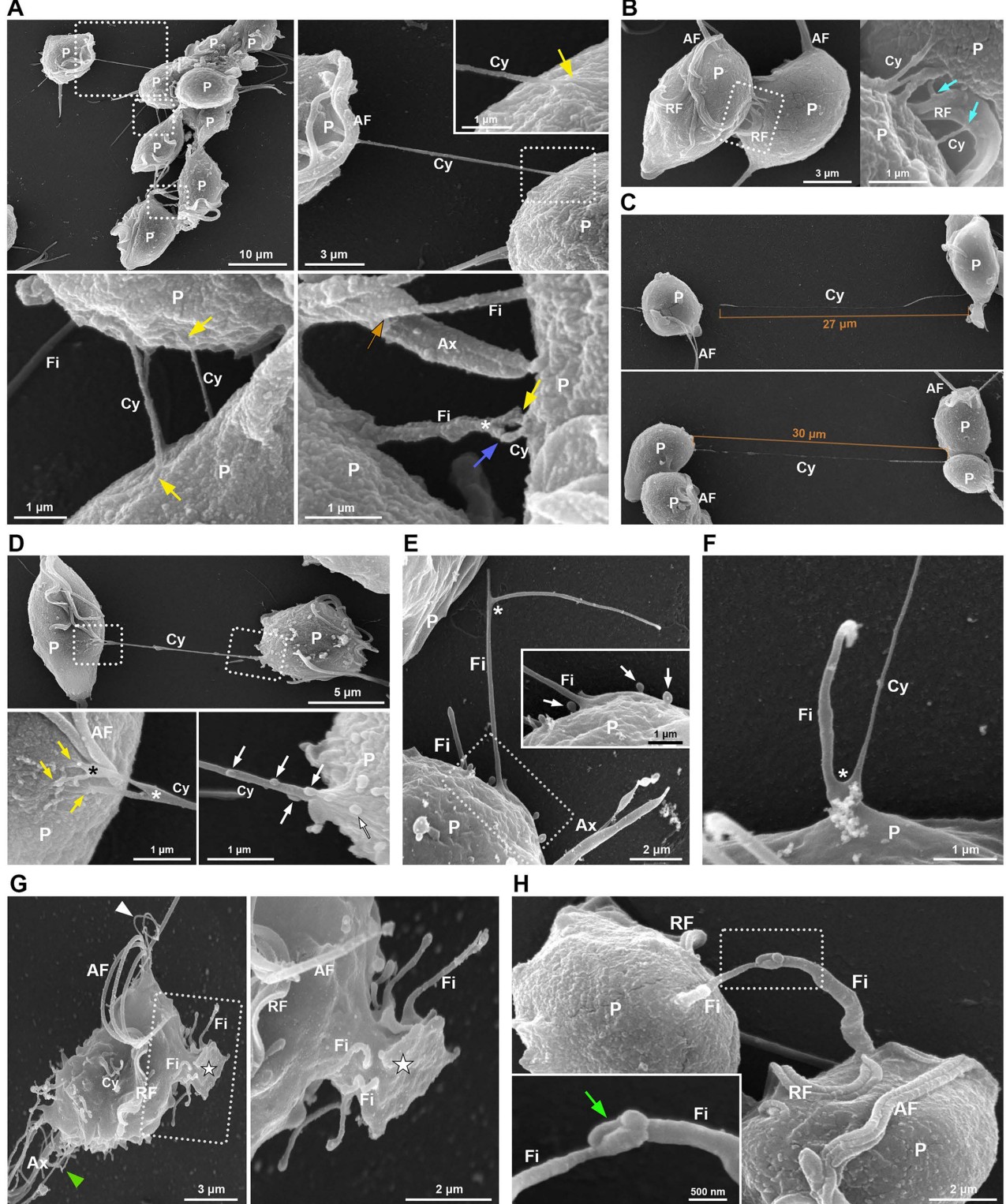

**Fig 3. SEM of cytonemes and filopodia in HMDS-dried *T. vaginalis* FMV1 strain during parasite-to-parasite interactions in axenic culture.** (A) General and detailed views of a parasite clump. Cytonemes (Cy) and filopodia (Fi) emerging from anterior flagella (AF) and cell body (P) are observed connecting adjacent cells within the clump of parasites. Points of contact between cytonemes and the cell body, axostyle (Ax), and filopodia are indicated

by yellow, orange, and dark blue arrows, respectively. The asterisk indicates a filopodium, which bifurcates. (B) Cytonemes (Cy) are seen contacting the surface of the recurrent flagellum (RF) of an adjacent parasite (light blue arrows). (C) Long cytonemes (Cy) are observed extending toward (upper image) and establishing contact with (lower image) a neighboring parasite (P). (D) General and detailed views of a cytoneme (Cy) connecting two parasites (P). The distal end of the projection is bifurcated (*) and is contacting the cell body of the adjacent parasite (yellow arrows). Microvesicle-like structures are seen on the proximal region of the cytoneme (white arrows). (E) A bifurcated filopodium (Fi) protruding from the parasite cell body (P) is shown. Inset: microvesicles are seen emerging near the base of the filopodium (white arrows). (F) A filopodium (Fi) and a cytoneme (Cy) emerge from a common bifurcation (*) on the parasite surface (P). (G) General and detailed views of multiple filopodia (Fi) protruding from a pseudopod-like structure (★). Flagellar cytonemes and posterior projections are indicated by white and green arrowheads, respectively. (H) General and detailed views showing the interaction between the distal ends of two filopodia (Fi) from adjacent parasites (P), forming a handshake-like contact structure (green arrow).

we observed no surface projections within clumps of CPD-dried samples (S6 Fig), suggesting that the visualization of these structures is facilitated by the use of HMDS drying.

Interestingly, SEM analysis of HMDS-dried FMV1 parasites also revealed that some cytonemes and filopodia exhibited branching, forming one or more bifurcations that can contact multiple regions of adjacent cells (Fig 3A, 3D–E). Occasionally, a filopodium and cytoneme were observed emerging from a shared bifurcation point (Fig 3F). Additionally, multiple filopodia were seen projecting from a single pseudopod-like structure in some individual parasites (Fig 3G). Similar bifurcated cytonemes and filopodia have been described in *Dictyostelium*, *Drosophila*, and zebrafish, where they are implicated in cell-to-cell communication, particularly in the transfer of signaling molecules during development and tissue maintenance [69–72]. The formation of such bifurcations has been previously associated with the activity of formins, a family of actin-filament nucleator proteins [70,73]. The *T. vaginalis* genome encodes several canonical actin-associated proteins, including α-actinins, capZ, cofilin, fimbrin, profilin, the Arp2/3 complex, and formin [19,74]; however, their specific roles in the parasite's biology remain poorly understood. Given the significance of these protein families in filopodium and cytonemes formation, it would be valuable to conduct a comprehensive investigation of each of them. Given the established importance of these proteins in the formation of filopodia and cytonemes, further investigation into their function in *T. vaginalis* is warranted.

Studies in *Drosophila* and zebrafish have demonstrated that cytonemes can transfer extracellular vesicles (EVs) along their surface in a highly directional manner [64,75–77]. These vesicles play key roles in intercellular communication by carrying signaling molecules that are subsequently delivered to recipient cells [75–77]. Interestingly, we previously reported that strain-specific EVs from *T. vaginalis*, which contain actin-associated proteins, can stimulate the formation of filopodia and cytonemes in recipient parasites, suggesting potential synergistic mechanisms involving paracrine signaling and cytoneme-mediated communication [9]. However, in that earlier study, we were unable to visualize a direct association between vesicles and cytonemes using SEM of CPD-dried parasites [9]. In contrast, in the present study, using the HMDS-based protocol, microvesicle-like structures were occasionally observed along the surface of cytonemes and filopodia, preferably localized to their proximal regions (Fig 3D–E). This finding raises the possibility that signaling molecules might be transported via vesicles along these projections; however, further studies are needed to confirm this hypothesis and elucidate the underlying mechanisms.

Axial twisting and rotation are common behaviors of cellular filopodia, arising naturally from the torsional forces generated by the spinning of the actin shaft. These dynamics can lead to a range of physical phenomena, including tip movement, helical buckling, and coiling [78]. Consistent with this, our previous CPD-based analysis demonstrated that filopodia, cytonemes, and other surface projections of *T. vaginalis* frequently exhibit helical buckling and coiling, suggesting that a similar actin shaft over-twisting mechanism may be involved [9,14]. In the present study, similar morphological features were observed in HMDS-dried parasites (Fig 3A–D, 3F–H). Moreover, we also identified interactions between the distal ends of two filopodia from adjacent parasites, forming a handshake-like contact structure (Fig 3H). This configuration resembles the interdigitated filopodia observed at adherens junctions between epithelial cells in metazoans [79].

**CPD and HMDS drying exhibit similar performance in preserving microvesicle-like structures in *T. vaginalis* surface**

Studies have demonstrated that *T. vaginalis* releases extracellular vesicles (EVs), including microvesicles, from both the cell body and flagellar surface under axenic culture conditions [15,17,80]. These vesicles carry strain-specific cargo involved in various biological processes and are implicated in parasite–parasite and parasite–host interactions, including the modulation of cytoneme and filopodia formation, parasite adhesion to epithelial cells, and host immune responses [9,15,80,81]. Here, we evaluated and compared the effects of CPD and HMDS drying procedures on the ultrastructural preservation of microvesicles on the surface of *T. vaginalis* FMV1 strain under axenic growth conditions (Fig 4).

SEM analyses showed that the CPD and HMDS-based protocols yielded comparable results regarding the ultrastructural preservation of microvesicle-like structures on the surface of the parasites. Numerous microvesicles were often observed budding from the cell body of FMV1 parasites processed by either method (Fig 4A). To confirm this, we quantified the percentage of parasites displaying microvesicles on their surface upon CPD and HMDS drying (Fig 4B). Similar percentages were observed for both methods, with 42% and 43% of FMV1 parasites exhibiting microvesicles after CPD and HMDS procedures, respectively. Additionally, consistent with the previous studies [15,17], microvesicles were seen emerging from the anterior flagella in both CPD- and HMDS-dried parasites (Fig 4C). As expected, cytonemes and filopodia were frequently observed in HMDS-dried samples (Fig 4A and 4C).

Comparable results were also obtained for both the CDC1132 and G3 strains. In CDC1132, microvesicles were observed in 38% and 33% of parasites processed by CPD and HMDS, respectively. In the G3 strain, approximately 10% of parasites exhibited microvesicles under both drying conditions (S1 Dataset). These findings suggest that, unlike filopodia and cytonemes, microvesicles are more resistant to the physical stresses associated with CPD and are consistently preserved in all three strains. Moreover, our data supports the suitability of HMDS as a reliable alternative to CPD for the ultrastructural visualization of *T. vaginalis* microvesicles by SEM. Similarly, HMDS-based drying has been successfully employed for the preservation of extracellular vesicles budding from the cell surface in other protists, including *Naegleria fowleri* [43], *Leishmania infantum* [50], *Dunaliella tertiolecta* and *Phaeodactylum tricornutum* [42].

**HMDS is more effective than CPD in preserving the ultrastructure of the amoeboid *T. vaginalis* surface**

Another distinctive feature of highly adherent strains of *Trichomonas vaginalis* is their rapid amoeboid transformation upon contact with host cells or inert surfaces. The free-swimming, pear-shaped flagellated cell flattens and spreads across the substrate, adopting an amoeboid morphology characterized by numerous pseudopod- and filopodia-like projections [8,9,18,20,82–84]. This type of morphogenesis is uncommon among flagellated protists and, beyond the trichomonads, has only been reported in a few species, including *Naegleria* sp. [85,86] and *Physarum polycephalum* [87]. Although *T. vaginalis* can spontaneously adhere to glass and plastic surfaces, a higher proportion of amoeboid adherent cells is observed when the substrate is coated with fibronectin [82].

To assess whether, as observed with free-swimming parasites, HMDS better preserves amoeboid *T. vaginalis* morphology, FMV1 and CDC1132 strains were incubated on fibronectin-coated coverslips and processed for SEM using both CPD and HMDS (Figs 5, S7 and S8). Samples were obtained from the same culture tube and subjected to identical fixation, post-fixation, dehydration, and handling conditions. In CPD-dried FMV1 samples, surface artifacts such as retracted lamellipodia and ruptured lamellipodia/pseudopods were frequent, and cytonemes connecting adjacent parasites were absent (Fig 5A). In contrast, HMDS-dried samples displayed numerous long and often bifurcated cytonemes forming extensive networks connecting multiple parasites over long distances. Although occasional ruptured projections were observed, most parasites exhibited well-preserved, flattened lamellipodia and pseudopods spread across the substrate (Fig 5A). Similar results were found for CDC1132 parasites (S7 Fig).

Quantitative analysis supported these observations. Approximately 91% of HMDS-dried amoeboid parasites exhibited surface projections, compared to only 29% in CPD-treated samples. Among the CPD-treated parasites with projections,

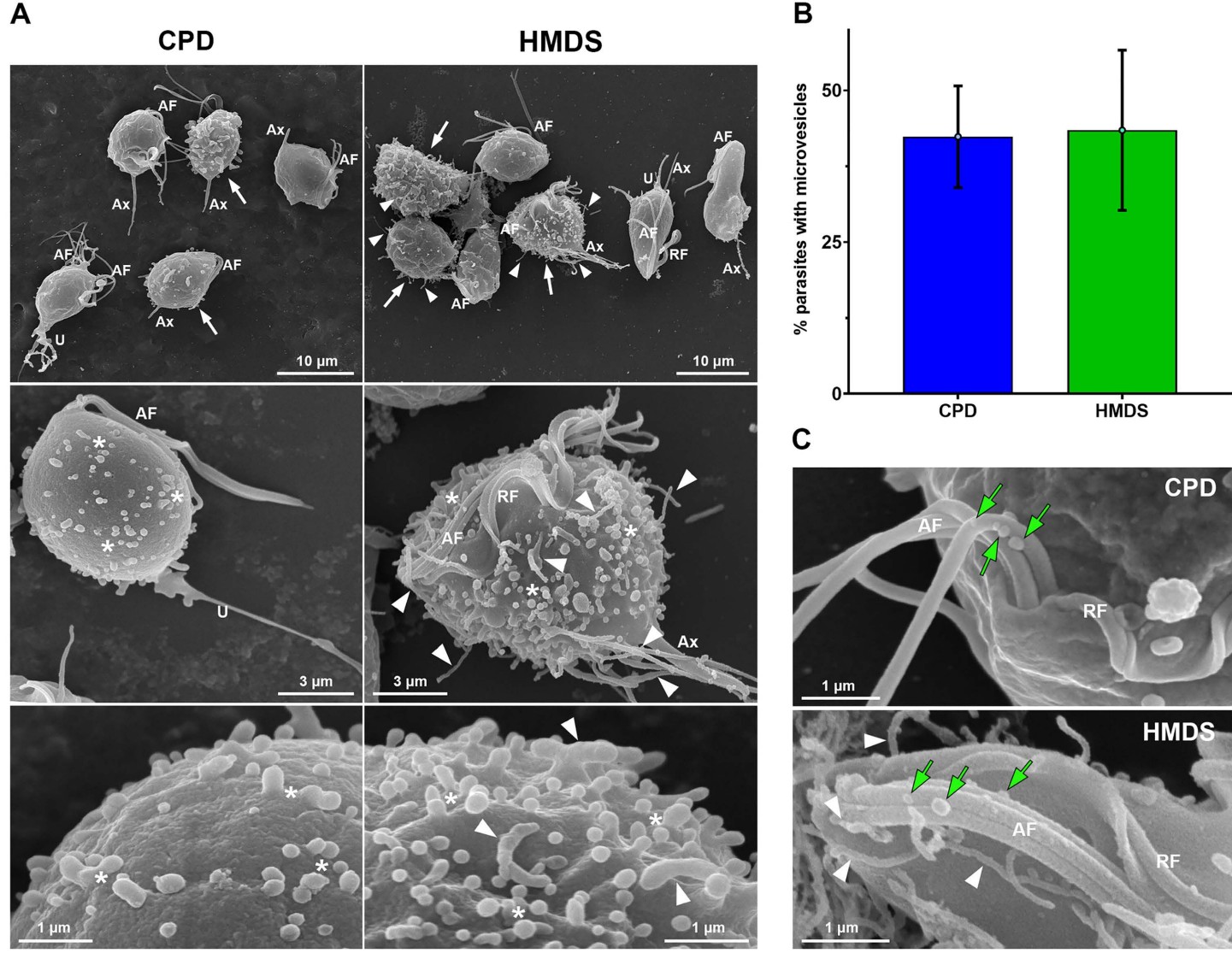

**Fig 4. Comparative analysis of CPD and HMDS drying reveals similar efficiency in preserving microvesicle-like structures on the surface of *T. vaginalis* FMV1 strain.** (A) Representative SEM images of general and detailed views of the *T. vaginalis* processed by CPD and HMDS drying. Numerous microvesicles (*) are seen protruding from the cell body of parasites processed by both drying methods (white arrows). Cytonemes and filopodia are observed in HMDS-dried parasites (arrowheads). (B) Quantification of the percentage of parasites exhibiting microvesicles on their cell surface upon CPD and HMDS drying. Bars represent the mean ± standard deviation from three independent experiments performed in duplicate. 500 parasites were randomly counted per sample using SEM. (C) SEM of microvesicles shedding from the anterior flagella (AF) of the parasites processed by CPD and HMDS drying. Cytonemes are observed in HMDS-dried parasites (arrowheads). RF, recurrent flagellum; Ax, axostyle; U, uropod-like projection.

two-thirds showed disrupted structures, whereas only one-fifth of HMDS-dried parasites exhibited similar damage (Fig 5B). These findings highlight the superior preservation offered by HMDS and suggest that adherent amoeboid parasites may establish long-distance communication through networks of cytoneme-like membranous connections, as previously observed in the adherent form of *Naegleria fowleri* [43].

Additionally, mesh-like structures were often observed on the cell surface of HMDS-dried FMV1 parasites, but not in CPD-treated samples. In contrast, CDC1132 parasites exhibited such structures under both drying methods (S8 Fig),

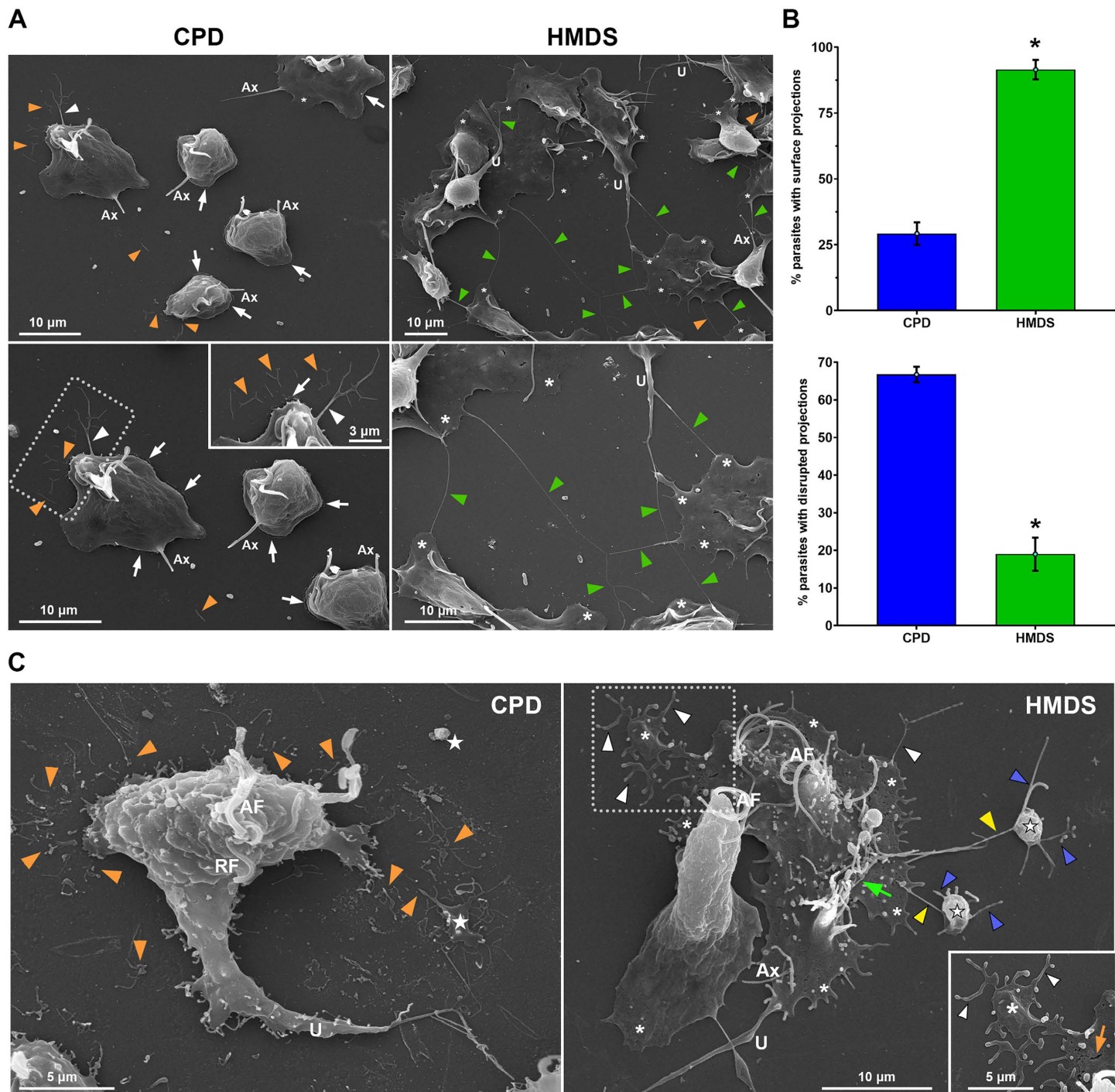

**Fig 5. HMDS provides superior ultrastructural preservation of the amoeboid *T. vaginalis* surface (FMV1 strain) following adhesion to fibronectin-coated coverslips.** (A) Representative SEM images showing general and detailed views of parasites processed by CPD and HMDS drying. In CPD-dried sample, lamellipodia and pseudopods are retracted (white arrows), and disrupted filopodia and cytonemes are evident (orange arrowheads). An intact surface projection is indicated by a white arrowhead. No cytonemes are observed connecting adjacent parasites. In contrast, HMDS-dried sample display well-preserved, long cytonemes forming a network between adjacent parasites (white arrowheads), with few ruptured projections (orange arrowheads). Large, flattened lamellipodia and pseudopods are observed spreading across the substrate (*). (B) Quantification of the percentage of parasites displaying filopodia and/or cytonemes (top graph) and the percentage exhibiting disrupted surface projections (bottom graph) following CPD and HMDS drying. Bars represent mean±standard deviation from three independent experiments performed in duplicate. 500 parasites

were randomly counted per sample using SEM. *p < 0.05, Mann–Whitney U test. (C) Detailed SEM images highlighting differences in surface preservation between CPD- and HMDS-dried parasites. In CPD samples, numerous fragments resembling ruptured filopodia, cytonemes (orange arrowheads), lamellipodia, and blebs (★) are seen around a parasite. In HMDS, intact filopodia and cytonemes (white arrowheads) are seen extending from pseudopods and lamellipodia (*) of a parasite. Filopodia are also observed connecting lamellipodia (*) to large blebs (★), forming network-like arrangements (yellow arrowheads). Filopodia protruding from blebs are indicated by blue arrowheads. A cluster of projections at the posterior region of the parasites is indicated by a green arrow. Minor surface ruptures are indicated by an orange arrow. Ax, axostyle; U, uropod-like projection.

supporting our findings that surface projections in the FMV1 strain are more vulnerable to CPD-induced damage than those of the CDC1132 strain. These formations may represent membrane reserves used during rapid amoeboid transformation, consistent with the need for surface area expansion during morphological changes in *T. vaginalis*. To facilitate rapid changes in morphology without compromising cell integrity, the cell possesses a substantial amount of structures called as cell surface excess. Such structures can be stored in different types of small surface projections to be promptly deployed to cover cell extensions [88].

Previous studies have shown that the migration of amoeboid forms of *T. vaginalis* is primarily driven by cell protrusions such as pseudopods, lamellipodia, and filopodia, rather than by bleb-based motility—a characteristic mode of movement observed in other amoeboid cells, including protists like *Entamoeba* and *Naegleria*, as well as in cancer cells [8,20,84,89–92]. However, the structural organization and the mechanisms underlying protrusion formation during *T. vaginalis* amoeboid transformation at the ultrastructural level remain poorly understood, partly due to the difficulty of preserving these delicate structures during sample processing. In CPD-dried samples, parasites were frequently observed surrounded by numerous fragmented structures resembling ruptured filopodia, cytonemes, lamellipodia, and blebs (Fig 5C), though their origin—parasite-derived or debris—could not always be determined.

In contrast, HMDS-dried parasites exhibited markedly improved surface preservation. A robust formation of intact filopodia and cytonemes was frequently observed emerging from lamellipodia and pseudopods (Fig 5C). Clusters of projections were occasionally observed at the posterior region, and filopodia were seen linking lamellipodia to bleb-like structures, forming a network-like arrangement not previously described in *T. vaginalis*. Multiple filopodia were also observed extending from these blebs (Fig 5C). The morphology of these interconnected structures closely resembles membrane protrusions described during the migration of *Theileria annulata*-infected macrophages. In that context, Ma and Baumgartner characterized a novel mode of amoeboid migration involving the coordinated action of actin-driven filopodia and pressure-induced blebs, in which blebs expand along filopodia-like projections, initiating at the base and extending outward [91]. Such blebs are frequently decorated with multiple filopodia [91].

Based on our SEM observations, it is tempting to hypothesize that a similar mechanism might occur during amoeboid migration in *T. vaginalis*. Although further investigations using molecular and live-cell imaging approaches are necessary to confirm this, our data highlights the potential of HMDS drying to preserve delicate surface projections, offering valuable ultrastructural insights into the morphological plasticity and potential migratory behavior of *T. vaginalis*.

## HMDS improves visualization of *T. vaginalis* on exposure to host cells

Next, to evaluate the effectiveness of the HMDS protocol in preserving the morphology of *T. vaginalis* during adhesion to host cells, FMV1 and CDC1132 strains were co-incubated with BPH1 prostate epithelial cells and processed for SEM using both CPD and HMDS (Fig 6). Samples were derived from the same culture tube and subjected to identical fixation, post-fixation, dehydration, and handling conditions. Consistent with our observations in axenic culture, CPD-dried FMV1 parasites exhibited only uropods and occasional small protuberances on the cell body, with few or no filopodia or cytonemes after contact with BPH1 cells. In contrast, HMDS-dried FMV1 parasites displayed an abundance of surface projections, including numerous filopodia and cytonemes establishing contact with adjacent parasites, host cells and bacteria (Fig 6A). For CDC1132, in both drying methods, we frequently observed multiple filopodia and cytonemes protruding

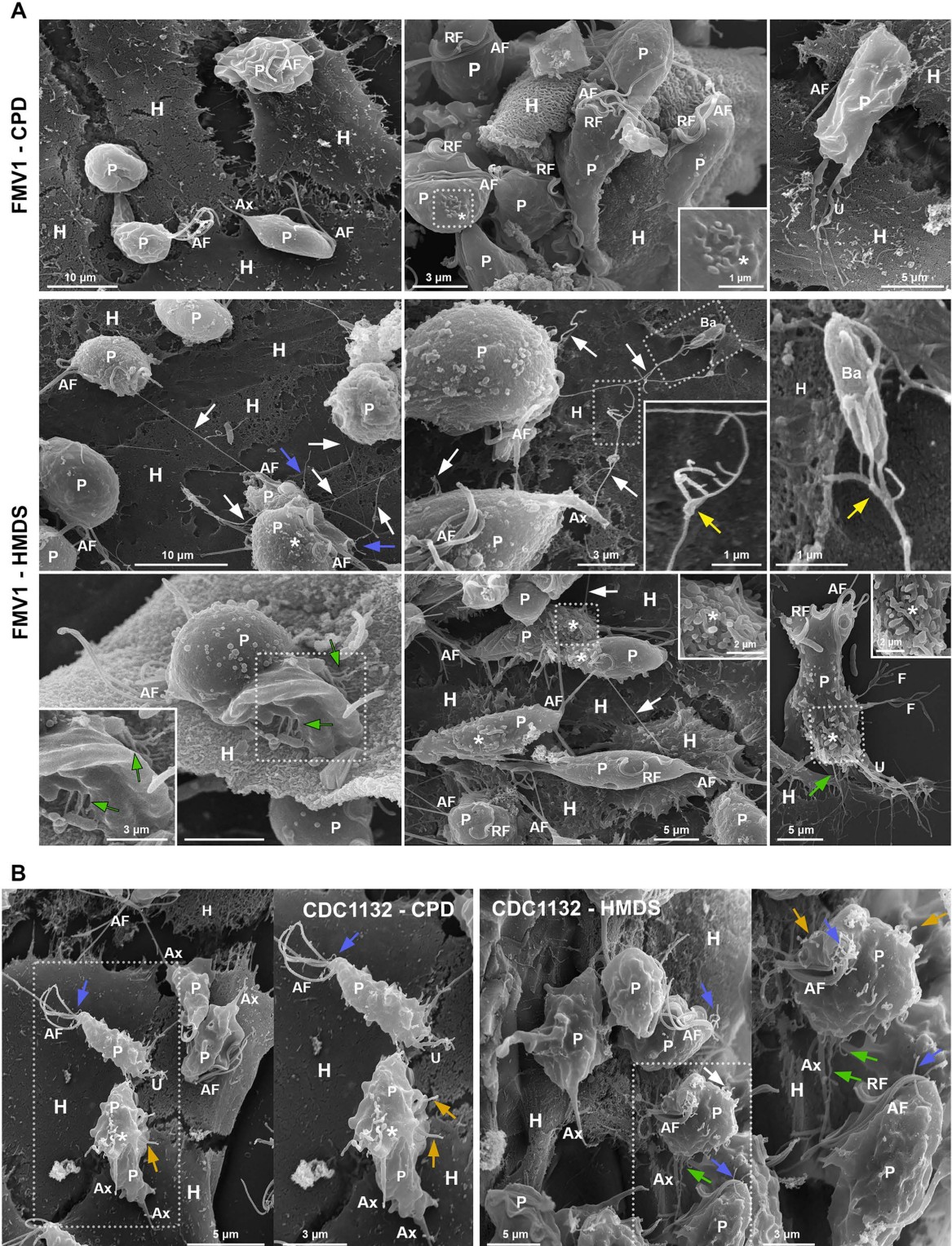

**Fig 6. HMDS enhances ultrastructural preservation of *T. vaginalis* during interaction with BPH1 prostate epithelial cells.** Representative SEM images showing general and detailed views of parasites (P) in contact with hosts cells (H) following by CPD and HMDS drying. (A) FMV1 strain. In CPD-dried sample (top row), no filopodia or cytonemes are observed on the surface of the parasites after contact with BPH1 cells. A cluster of small surface

protuberances is observed on one parasite (*, inset), and a uropod-like projection (U) is seen contacting a host cell. In contrast, HMDS-dried samples (middle and bottom rows) reveal long cytonemes (white arrows) in contact with neighboring parasites, host cells, and a bacterium (Ba). Flagellar and bifurcated cytonemes are indicated by blue and yellow arrows, respectively. Posterior and axostylar cytonemes extending toward host cells are indicated by green arrows. Clusters of protrusions at the posterior region of parasites are indicated by asterisks (insets), and the letter (F) denotes filopodium. (B) CDC1132 strain. In both CPD- and HMDS-dried samples, filopodia and cytonemes are observed emerging from the flagellar base (blue arrows) and cell body (orange arrows) of parasites interacting with BPH1 cells. A cluster of projections at the posterior region is indicated by an asterisk. Elongated posterior and axostylar cytonemes are seen in HMDS-dried sample (green arrow). AF, anterior flagella; Ax, axostyle; RF, recurrent flagellum.

from both the flagellar base and cell body of parasites in contact with BPH1 cells, as previously shown [9]; however, HMDS-dried samples revealed a higher number of elongated posterior and axostylar cytonemes, suggesting improved preservation of these structures (Fig 6B).

HMDS drying revealed numerous long, often bifurcated cytonemes forming networks that connect multiple parasites during host cell interaction (Fig 6A). Although we recently showed that cytonemes have a role in the process of *T. vaginalis* attachment to prostate cells, we were unable to find cytoneme connections between parasites following host cell adhesion using SEM with CPD drying [9]. Therefore, our present findings suggest that cytoneme-like structures may mediate parasite-to-parasite communication during host interaction.

Additionally, HMDS-dried samples revealed numerous previously undescribed posterior and axostylar cytonemes mediating *T. vaginalis* adhesion to BPH-1 cells (Fig 6). These structures resemble the uroid filaments of *Naegleria fowleri*, which extend from the posterior region and facilitate attachment to mammalian cells or other substrates [43]. Consistent with our findings, previous studies have demonstrated that *T. vaginalis* employs various strategies to explore and anchor to host surfaces via its posterior region, including the use of the axostyle and uropod-like projections [8,14,88,93].

Moreover, we observed clusters of protrusions at the posterior pole of parasites adhered to both fibronectin-coated coverslips (S8 Fig) and BPH-1 cells (Fig 6), resembling previously described surface excess structures [88]. In other protozoan parasites, such as *Entamoeba histolytica*, similar posterior projections have been implicated in membrane replenishment [94]. Furthermore, in flagellated parasites, the posterior region has been associated with the removal of surface-bound ligands, a process that may contribute to immune evasion [95].

## Conclusions

This study establishes a reliable, fast and effective HMDS drying protocol as an alternative to CPD for the ultrastructural analysis of *T. vaginalis* by SEM. Our comparative analyses show that HMDS significantly improves the preservation of fragile surface projections, such as filopodia and cytonemes, that are important players in parasite-parasite and parasite-host interactions. Our data indicate that susceptibility to CPD-associated artifacts can vary among highly adherent *T. vaginalis* strains, and this needs to be considered during SEM interpretation. We therefore strongly recommend the complementary use of both CPD and HMDS drying procedures whenever feasible, to ensure a more accurate, comprehensive, and artifact-minimized characterization of *T. vaginalis* surface morphology.

This study demonstrates that HMDS drying is not only a technically and economically viable protocol but also a valuable tool that enabled the identification of novel ultrastructural features of *T. vaginalis*, including: (a) Long branched cytonemes forming network-like structures between parasites and host cells; (b) Posterior and axostylar cytonemes involved in host cell adhesion; (c) Membrane surface excess-like structures; and (d) Filopodia linking lamellipodia to bleb-like structures. These findings provide new ultrastructural insights into the surface morphology and intercellular communication mechanisms of *T. vaginalis*.

However, HMDS is not a glitch-free method. It reduces the drying-associated artifacts but does not eliminate them. Moreover, if improperly handled, it may introduce its own artifacts. Therefore, careful protocol optimization and

comparison with other complementary preparative techniques remain essential to ensure the reliability and reproducibility of SEM data. Altogether, this study affirms the importance of the selected drying methods for SEM sample preparation and expands the methodological tool for ultrastructural preservation and *T. vaginalis* imaging.

## Materials and methods

### Parasites and cell cultures

*T. vaginalis* strains FMV1 [52], CDC1132 (MSA1132; Mt. Dora, Fla, USA 2008) and G3 (ATCC PRA-98; Beckenham, UK) were cultured in Diamond's Trypticase-yeast extract-maltose (TYM) medium supplemented with 10% fetal bovine serum and 10 U/ml penicillin/10 µg/ml streptomycin (Invitrogen). Parasites were grown at 37°C and passaged daily. The human BPH-1 cells, kindly provided by Dr Natalia de Miguel (Instituto Tecnológico Chascomús, Argentina) [96], were grown in RPMI 1640 medium containing 10% fetal bovine serum (Invitrogen) with 10 U/ml penicillin and cultured at 37°C under 5% $CO_2$.

### SEM

Parasites in suspension were harvested by centrifugation, washed with PBS, pH 7.2, and fixed in 2.5% glutaraldehyde in 0.1 M cacodylate buffer, pH 7.2. Following fixation, the parasites were rinsed in PBS, allowed to adhere to poly-L-lysine-coated glass coverslips, and subsequently transferred to a 24-well plate. Parasites were then post-fixed for 15 min in 1% $OsO_4$ and dehydrated in ethanol series (7.5%, 15%, 30%, 50%, 70%, 90%, and three times in absolute alcohol for 10 minutes each). After dehydration, coverslips of each sample were divided into two groups: (a) CPD, in which the parasites were critical point-dried with liquid $CO_2$ using an automated LEICA EM CPD300 system under the following parameters: $CO_2$ IN Speed – slow, delay 10 min; Combination of 1/2 and 1/3 holders; Exchange speed 5 (medium flowing speed); Exchange cycles – 15 cycles; Heating speed – slow; Gas out speed – slow 50%; and (b) Chemical drying, in which several protocols using HMDS (Sigma-Aldrich, catalog numbers 440191 and 379212) were tested, as shown in S1–S3 Figs. In the optimized protocol (Fig 1A and S1 Movie), samples were sequentially incubated in HMDS: ethanol 1:2, 1:1 and 2:1 mixture (final volume 500 µL) for 10 min each, followed by a brief immersion (up to 30 seconds) in 100% HMDS (300 µL). Samples were then blotted on filter paper and air-dried for 30 minutes. In some assays, aliquots were directly air dried after ethanol dehydration. All dried parasites were coated with gold–palladium to a thickness of 15 nm and then observed with a Jeol JSM-5600 scanning electron microscope, operating at 15 kV.

### Fibronectin- parasite adhesion assay

Fibronectin-coated coverslips were prepared by applying 100 µL of human fibronectin (Sigma F0556; 10 µg/mL in sterile PBS) for 1 hour at room temperature, followed by rinsing with sterile PBS. Parasites ($1 \times 10^6$ cells/mL) were washed and resuspended in PBS (pH 7.2). A 50 µL aliquot of the parasite suspension was incubated on the fibronectin-coated coverslips in a humid chamber for 30 minutes at 37°C. Parasite adhesion was monitored using an inverted phase-contrast microscope. Subsequently, coverslips were thoroughly washed with PBS to remove non-adherent cells. Adherent parasites were then fixed and processed for SEM analysis as described above.

### Parasite–host cell interaction

BPH1 cells were washed twice with pre-warmed PBS by centrifugation at $400 \times g$ for 5 min and resuspended at a concentration of $1 \times 10^5$ cells/mL in warm PBS. The cells were then co-incubated with *T. vaginalis* at a parasite-to-host cell ratio of 5:1 in PBS-F (PBS supplemented with 1% fetal bovine serum, pH 6.5) at 37°C for 30 min. Prior to co-incubation, *T. vaginalis* were washed three times in PBS (pH 7.2) and incubated in PBS-F at 37°C for 15 min. After co-incubation, samples were fixed and processed for SEM as described above.

## Quantitative and statistical analysis

All results represent the mean of three independent experiments; each performed at least in duplicate. Surface area measurements of CPD- and HMDS-dried parasites were obtained from SEM images of at least 50 randomly selected parasites per sample using ImageJ software (https://imagej.nih.gov/ij/download.html), employing the "area" parameter. The percentage of parasites exhibiting surface projections (cytonemes/filopodia), disrupted projections, and microvesicles was determined by analysing SEM images of a minimum of 500 randomly selected parasites per sample. The diameter and length of cytoneme-like projections in HMDS-dried FMV1 strain parasites were measured from SEM images of 150 randomly selected projections using the "perimeter," "Feret's diameter," and "length" parameters in ImageJ. Statistical comparisons were performed using the Mann–Whitney U test in GraphPad Prism software (v. 9.5.0; GraphPad Software, CA, USA). A $p$-value $< 0.05$ was considered statistically significant.

## Supporting information

**S1 Fig. Protocol 1 – Schematic representation of sample preparation and representative SEM images of *T. vaginalis* FMV1 strain processed by CPD and HMDS drying.** In CPD-prepared samples, parasites exhibit typical piriform or ellipsoid morphology with a slightly irregular surface displaying small undulations. Four anterior flagella (AF), a recurrent flagellum (RF), and the axostyle (Ax) tip extending from the posterior region of the parasite are seen. Artifacts such as cell shrinkage (white arrows) and ruptured structures (blue arrows) are also observed. In HMDS-prepared samples, parasites display a flattened surface; anterior flagella (AF), the undulating membrane (black asterisks), and axostyle (Ax) are not clearly distinguishable and appear adhered to the substrate (white arrowheads). A white asterisk indicates a parasite covered with HMDS film, resulting in the fusion of individual structures into a single continuous mass.
(JPG)

**S2 Fig. Protocol 2 – Schematic representation of sample preparation and representative SEM images of general and detailed views of *T. vaginalis* FMV1 strain processed by CPD and HMDS drying.** In CPD-prepared samples, parasites exhibit typical piriform or ellipsoid morphology with a slightly irregular surface displaying small undulations and microvesicles-like structures. Four anterior flagella (AF), a recurrent flagellum (RF), and the axostyle (Ax) tip in the posterior region of the parasite are clearly observed. In HMDS-prepared samples, parasites appear coated with a thick HMDS film, resulting in a wrinkled and shrunken surface and fusion of individual parasites into a single continuous mass, with a concomitant loss of distinct morphological features (*).
(JPG)

**S3 Fig. Protocol 3 – Schematic representation of sample preparation and representative SEM images of *T. vaginalis* FMV1 strain processed by CPD and HMDS drying.** In CPD-prepared samples, parasites displaying typical piriform or ellipsoid morphology with a slightly irregular surface displaying small undulations and microvesicles-like structures. Four anterior flagella (AF), a recurrent flagellum (RF), and the axostyle (Ax) tip in the posterior region of the parasite are clearly observed. In HMDS-prepared samples, parasites appear coated with a thick HMDS film, resulting in a wrinkled and shrunken surface, and fusion of individual parasites into a single continuous mass, accompanied by the loss of distinct morphological features (*).
(JPG)

**S4 Fig. Representative SEM images of *T. vaginalis* FMV1 strain processed by air drying.** Morphological damage is evident in air-dried samples, including cell shrinkage, disrupted flagella (arrows), surface ruptures (arrowheads), and the presence of a perforation (*). AF, anterior flagella; RF, recurrent flagellum.
(JPG)

**S5 Fig. SEM images of *T. vaginalis* FMV1 strain processed by CPD.** Worm-like structures (arrows) are observed on the cell body and at the flagellar base region of the parasites. AF, anterior flagella; RF, recurrent flagellum.
(JPG)

**S6 Fig. SEM images of clumps of CPD-*T. vaginalis* FMV1 strain grown in suspension in axenic culture.** No surface projections are observed within the clumps. AF, anterior flagella; RF, recurrent flagellum.
(JPG)

**S7 Fig. Representative SEM images of CDC1132 parasites processed by CPD and HMDS drying after adhesion to fibronectin-coated coverslips.** General and detailed views show notable differences in surface preservation between the two drying methods. In CPD-dried sample, lamellipodia and pseudopods appear retracted (white arrows), ruptured filopodia and cytonemes are observed (orange arrowheads) and fragments are seen (★). In contrast, HMDS-dried sample exhibit long, intact cytonemes connecting adjacent parasites, forming an intercellular network (white arrowheads). Broad, well-preserved lamellipodia and pseudopods are also seen spread across the substrate (*).
(JPG)

**S8 Fig. SEM images of *T. vaginalis* following adhesion to fibronectin-coated coverslips.** (A) HMDS-dried FMV1 parasites. (B) CPD- and HMDS-dried CDC1132 parasites. Mesh-like structures on the cell body are indicated by arrows. In (A), the inset shows cytonemes protruding from the cell body (arrowheads). In (B), pseudopods are indicated by asterisks. AF, anterior flagella; RF, recurrent flagellum.
(JPG)

**S1 Dataset. The values behind the means, standard deviations and other measures reported, as well the values used to build the graphs.**
(DOCX)

**S1 Movie. Preparing *T. vaginalis* for SEM by HMDS drying.**
(MP4)

## Acknowledgments

We thank Dr. Marlene Benchimol from Universidade do Grande Rio for kindly providing *T. vaginalis* FMV1 strain. We thank Dr. Natalia de Miguel from Instituto tecnologico de Chascomus for kindly providing BPH1 cells and *T. vaginalis* G3 and CDC1132 strains. We also thank Dr Karina Saraiva and Dr Cássia Docena from the Technological Platform Core of the Aggeu Magalhães Institute for their technical support.

## Author contributions

**Conceptualization:** Regina Celia Bressan Queiroz de Figueiredo, Antonio Pereira-Neves.

**Formal analysis:** Tuanne dos Santos Melo, Abigail Miranda-Magalhães, Regina Celia Bressan Queiroz de Figueiredo, Antonio Pereira-Neves.

**Funding acquisition:** Regina Celia Bressan Queiroz de Figueiredo, Antonio Pereira-Neves.

**Investigation:** Tuanne dos Santos Melo, Abigail Miranda-Magalhães, Regina Celia Bressan Queiroz de Figueiredo, Antonio Pereira-Neves.

**Methodology:** Tuanne dos Santos Melo, Abigail Miranda-Magalhães, Regina Celia Bressan Queiroz de Figueiredo, Antonio Pereira-Neves.

**Project administration:** Regina Celia Bressan Queiroz de Figueiredo, Antonio Pereira-Neves.

**Resources:** Regina Celia Bressan Queiroz de Figueiredo, Antonio Pereira-Neves.

**Supervision:** Regina Celia Bressan Queiroz de Figueiredo, Antonio Pereira-Neves.

**Visualization:** Tuanne dos Santos Melo, Abigail Miranda-Magalhães, Regina Celia Bressan Queiroz de Figueiredo, Antonio Pereira-Neves.

**Writing – original draft:** Tuanne dos Santos Melo.

**Writing – review & editing:** Regina Celia Bressan Queiroz de Figueiredo, Antonio Pereira-Neves.

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
