## [Decision Letter · Decision Letter 0]

22 Aug 2025

We look forward to receiving your revised manuscript.

Kind regards,

António Machado

Academic Editor

PLOS ONE

“APN and RCBQF: grant from Fiocruz PROEP-IAM IAM-005-FIO-22-2-13

TSM and AMM are PhD fellow from Coordenação de Aperfeiçoamento de Pessoal de Nível Superior (CAPES)- Finance Code 001.”

3. We note that your Data Availability Statement is currently as follows: All relevant data are within the manuscript and in Supporting Information files.

Additional Editor Comments:

I am pleased to inform you that one reviewer already accepted the revised manuscript for publication and the other reviewer (reviewer 3) requested some revisions for future publication endorsement. Please carefully answer reviewer 3' concerns and rectify the manuscript following his/her comments.

Thank you for choosing PLOS ONE journal and best regards,

António Machado

Reviewers' comments:

Reviewer's Responses to Questions

**Comments to the Author**

1. Is the manuscript technically sound, and do the data support the conclusions?

Reviewer #1: Yes

Reviewer #2: No

Reviewer #3: Yes

2. Has the statistical analysis been performed appropriately and rigorously?

Reviewer #1: Yes

Reviewer #2: No

Reviewer #3: Yes

3. Have the authors made all data underlying the findings in their manuscript fully available?

Reviewer #1: Yes

Reviewer #2: Yes

Reviewer #3: Yes

4. Is the manuscript presented in an intelligible fashion and written in standard English?

Reviewer #1: Yes

Reviewer #2: No

Reviewer #3: Yes

Reviewer #1: Pereira-Neves et al present here a manuscript examining the utility of using HMDS over CPD techniques in SEM preparation and use this protocol demonstrate multiple novel and important findings in the surface structures of T. vaginalis. The work is original and insightful, thoroughly referenced, and very well written, both from a logical presentation standpoint as well as a readability angle.

First, of all, I have recently gotten in the habit of refusing to review most SEM-based research manuscripts: in most of the ones I've seen in the past few years the quality of the micrographics is technically poor and bordering on unacceptable, with incorrect astigmatism settings, charging artifacts, or other signs of frankly bad technical skills. Occasionally I can see that those authors have sought to fix these problems with post-processing (sharpness filters, etc.) - typically making the issue even worse. I am enormously pleased to say that this manuscript shows none of those problems! The images are clear and without distortion, with excellently managed depth of field (in particular the surface detail on the suspended filopodia and cytonemes in figure 3 are masterfully done). Charging artifacts are nearly zero (there are a couple of minor issues in the supplemental figures, but they do not interfere with interpretation and are entirely reasonable). Composition is also very well done, with meaningful labeling and unobtrusive scale bars noted throughout. Brilliant microscopic technique - just top notch.

Secondly, I've personally experimented rather extensively with HMDS in the past -- both in prokaryotic and eukaryotic applications -- but was always frustrated by the plethora of new artifacts the method exhibited in my hands. I can see now that I was doing it all wrong: none of my HMDS specimen exposure times were as short as this manuscript’s work demonstrates to be ideal. I will be revisiting this in light of these findings for my own research. Of note, I also appreciate the caution the authors note about presuming that one of the HMDS or CPD options are "best": specimen drying for SEM is a finicky beast and advocating for a testing both methods with new work is appropriate and demonstrates extensive thought (and experimentation!) by the researchers. An interesting extension of this work would be to use cryoSEM to examine the "native state" of the T. vaginalis surface-associated appendages without chemical or physical drying artifacts. (While this would be an excellent "future work" component, it is obviously beyond the scope of the current manuscript.)

Finally, the novel biological findings presented in the manuscript are accompanied by reasonable and thoughtful interpretations - the authors have avoided making dramatic claims even with wonderfully clear micrographs that are strongly suggestive.

The length of my reviews is generally inversely proportional to the quality of a manuscript - hence this review is short. I applaud the authors for one of the clearest, well supported, and well written papers I have reviewed in quite some time: kudos!

Reviewer #2: The authors have submitted a manuscript titled “Replacing critical point drying with hexamethyldisilazane drying enhances the ultrastructural preservation of cell surface projections in the parasite Trichomonas vaginalis for scanning electron microscopy”, presents hexamethyldisilazane (HMDS) drying as a reliable alternative to conventional sample preparation via critical point drying.

However, the study has serious limitations, particularly concerning the methodology followed and the overall study design. While the study attempts to present an intriguing narrative and appears to have generated some positive results, several significant concerns need to be addressed.

1. The current sample size is small, which limits the statistical reliability of the findings. The authors are advised to increase the sample size (consider different species and strains of Trichomonas) and repeat the experiments to ensure more robust and generalizable results.

2. The Materials and Methods section should be described in detail, with clear alignment to the results presented. Relevant protocols should be adequately cited, and sufficient information must be provided to allow reproducibility.

3. The Results and Discussion section should be rewritten to ensure clear presentation of findings, logical interpretation, and alignment with the study objectives.

Specific comments:

1) Fibronectin- parasite adhesion assay > Authors should provide proper citation for the method.

2) Parasite–host cell interaction > Authors should provide proper citation for the method.

3) Please mention what are the major difference between 3 HMDS protocols.

Reviewer #3: This paper by dos Santos Melo and colleagues was an interesting read. The differences between the present study’s approach and that of a previous study (Malli and colleagues 2018) is clearly discussed (as it should be, given the shared context between it and the present manuscript), although it is a bit odd this is primarily covered at the beginning of the results section rather than the Introduction. Actually, my impression was that the end of the Introduction itself could be improved regarding clarity and structure; the Introduction reasonably overviewed Trichomonas vaginalis, its established relevance as a sexually transmitted pathogen, and reasonably overviewed SEM/CPD and potential issues with it and why HMDS drying might be preferable (in a broad sense). However, the connection of these overviews to the core focus of the paper (application of HMDS incubation vs CPD to T. vaginalis compared against the results of the prior study by Malli et al. 2018) could have been more smoothly done towards the end of the Introduction. However, the beginning of the Results section resolves this issue quite well, so I don’t want to complain about it too much. As is, the paper’s writing and structuring works well enough for the interested reader to follow it. The results and discussion sections were particularly informative, and, in my opinion, adequate checks/controls were performed to ensure the observed differences in cell surface morphology did not originate from experimental artifacts. Testing not only the FMV1 strain but also the known highly adherent CDC1132 and known poorly adherent G3 strains with both the HMDS and CPD protocols was a good decision in experimental design to ensure the observed cytoneme-like structures and extracellular vesicles (including microvesicles) in the HMDS FMV1 strain cells were not induced via exposure to HMDS. Additionally, the hypothesized explanation by the authors for why the FMV1 strain cell morphology responded so differently to preparation via CPD vs HMDS incubation was interesting and certainly not implausible. The authors don’t stop here, however; they then go and apply their modified HMDS drying protocol to observe T. vaginalis inter-cell interactions via filipodia, particularly via the cytonemes. Again, very appropriate experimental design in my opinion, and it led to an enjoyable read of this manuscript. Actually, the authors apply their modified HMDS protocol to the study of T. vaginalis cells in quite a few ways. This was unexpected and I don’t think strictly necessary for the publication of this manuscript (honestly I think it could almost be two separate, smaller papers), but it’s certainly not a problem. Seeing the structures observed on the amoeboid forms of the pathogen, as well as with the interaction of the pathogen with host prostate cells was interesting and certainly helps support the broad application/robustness of their modified HMDS sample drying approach in application to the T. vaginalis pathogen. Also, to be clear, my expertise relevant to this manuscript is more so regarding the SEM side of things; I certainly have a background in molecular/cell biology and microbiology, but I do not regularly research microbes or pathogens.

I will note, it would be nice if the underlying mechanisms explaining why the cells responded the way they did to HMDS incubation were able to be presented. It’s certainly fine they aren’t, but as is, this manuscript leans more towards a microbiology-heavy paper rather than an SEM protocol optimization paper applied to a microbiological context. The paper title and the Introduction sections imply a bit more of an emphasis on the SEM protocol optimization side of things, so this was initially a bit unexpected. However, again, the paper does work well for what it is, and I would not be too concerned over this. The way it’s written, it’s very informative; almost like a review article containing an SEM sample preparation protocol optimization procedure within it.

The comments below are all relatively minor, although I will note they apply mostly to the Introduction. The writing of the Results and Discussion sections was generally more appropriate than that of the Introduction. There are places in the Introduction section where the authors do need to be more balanced/measured in the information they present:

Line 44 replace the word “confirms” with perhaps “demonstrates” or something else; “confirms” is too close to “proves”, and obviously nothing is 100% proven within empirical science

Line 51 What makes the pathogen overlooked?

Lines 61-62 Statement needs to be cited, don’t just blanket claim many aspects of the pathogen cell biology and pathogenicity remain unknown or understudied

Line 67 “undoubtedly a powerful tool” change to perhaps “useful tool” especially the word “undoubtedly”; I agree SEM is a useful tool for study cell structure, but more measured wording would be more appropriate and less polarizing

Line 76 point out that the need for some sort of sample processing is common for most biological specimens, so readers unfamiliar with biological SEM understand this isn’t strictly limited to T. vaginalis surface structure

Line 76 I would change wording to “it generally requires meticulous processing” to be more measured; there are cases in which essentially unprepared biological samples can observed with SEM; for example, see environmental SEM

Lines 76-80 environmental SEM does exist, with which fully hydrated specimens can be observed using SEM; thus water removal is not always necessary

Line 81 “by far” again, I would change the wording to be a bit more measured, maybe say “is generally the most commonly used method”

Line 87 change “Although CPD is considered the state-of-the-art drying method” to “Despite the wide usage of the CPD drying method” or something comparable; as is, saying CPD drying is state-of-the-art is a strong assumption, and I’m not sure every SEM expert would refer to it in that manner. I certainly wouldn’t refer to it in that manner.

Lines 96-98 see, here it’s stated that hexamethyldisilazane has been used before as an alternative to CPD, so again, referring to CPD as state-of-the-art is a very arguable statement

Line 92 change “longer” to “substantial”, the word “longer” implies CPD is being compared to something, and that isn’t the case yet at this point in the Introduction

Lines 94-95 use more measured wording here than “highly recommended”, such as “may be preferable”; you’ve presented the evidence for your argument, try to avoid strongly pushing the reader towards thinking a specific way; let the reader decide for themselves

Line 95 add on “such as those of the T. vaginalis cell surface.” or something comparable at the end of this sentence on line 95 to help relate this SEM theory to your target pathogen

Line 112 wording must be changed to “that some highly adherent strains” because the very next sentence states that some other researchers have observed mixed numbers of cytoneme-like structures in some highly adherent strains; so clearly elevated numbers of cytoneme-like structures have not yet been observed for all strains of T. vaginalis; without this change in wording, these sentences are confusing and read as if they contradict themselves

Lines 362-368 would be helpful if these definitions for “filipodia” and “cytonemes” in context of T. vaginalis were presented earlier in the manuscript, given these terms are used quite a bit prior to their being defined here

Line 668 maybe use “affirms” instead of “confirms”; again, “confirms” is too close to “proves”

**Do you want your identity to be public for this peer review?** For information about this choice, including consent withdrawal, please see our Privacy Policy

Reviewer #1: No

Reviewer #2: No

Reviewer #3: No

---

## [Author Response · Author response to Decision Letter 1]

29 Aug 2025

Dear Dr. Machado, Academic Editor of PLOS ONE

Initially, we would like to thank you for handling our manuscript and for your thoughtful update on the review process and the Reviewers for their valuable time and comments which definitely contributed to improve the manuscript. We are very pleased to learn that one of the reviewers has already endorsed our revised submission, and we greatly appreciate Reviewer 3’s constructive feedback.

As can be seen in response to reviewers file, we carefully addressed all of his/her concerns and revised the manuscript accordingly to ensure it meets the journal’s standards. The alterations are highlighted in the file with the track changes. We believe that the manuscript has had improvement.

Thank you once again for your efforts on our behalf, and for considering our work for publication in PLOS ONE.

---

## [Decision Letter · Decision Letter 1]

17 Sep 2025

Replacing critical point drying with hexamethyldisilazane drying enhances the ultrastructural preservation of cell surface projections in the parasite Trichomonas vaginalis for scanning electron microscopy

PONE-D-25-35035R1

Dear authors,

I am pleased to inform you that both reviewers enjoyed the manuscript very much and endorsed the revised manuscript for publication.

Thank you for choosing Plos ONE journal to publish your study.

Best regards,

António Machado

Reviewers' comments:

Reviewer's Responses to Questions

**Comments to the Author**

Reviewer #3: All comments have been addressed

2. Is the manuscript technically sound, and do the data support the conclusions?

Reviewer #3: Yes

3. Has the statistical analysis been performed appropriately and rigorously?

Reviewer #3: Yes

4. Have the authors made all data underlying the findings in their manuscript fully available?

Reviewer #3: Yes

5. Is the manuscript presented in an intelligible fashion and written in standard English?

Reviewer #3: Yes

Reviewer #3: The authors have done an excellent job in revising this manuscript. Particularly, their revisions elevated the Introduction section from being merely acceptable to what I would now consider to be outstanding. I can happily recommend publication.

The comments listed below are almost all very minor, mostly grammatical. These should all be able to be dealt with between the authors and the editor without the need for further peer review:

Line 51 changing “overlooked” to “neglected" wasn’t what I meant here, either term is fine; rather, there needs to be some sort of justification for why you are claiming T. vaginalis to be “overlooked” or “neglected”. For example, that justification you gave to me in your reply to my comment on this in your reviewer response letter, that is what I was talking about. If you can add in a citation supporting this claim of T. vaginalis being “overlooked” or “neglected”, that should solve this issue. I’ve heard of T. vaginalis before and microbiology isn’t my central area of research, so I was surprised to read it was “overlooked” or “neglected”

Line 126 space needed after word “filipodia”

Line 127 maybe add the word “cells” after “T. vaginalis” here

Line 146 maybe replace “revealed” with “suggest”; I don’t believe you directly show the surface structures are fragile to CPD drying, but it is certainly suggested given the differences observed in cell surface morphology between the two drying processes; you do reveal CPD causes artifacts regarding the loss or absence of surface structure in the FMV1 strain relative to using the HMDS drying method

Line 192 check “cytonemes-like” for typo if it needs to be “cytoneme-like”

Line 235 change “150nm” to “150 nm”

Line 282 capitalize “representative”

Line 283 capitalize the “a” in “a myriad of surface projections” so it is “A myriad of surface projections”

Line 295 capitalize “representative”

Line 396 space needed between the period and (E)

Line 414 “FMV1parasites” should be “FMV1 parasites”

Line 416 change “Occasionally, filipodium and cytoneme” to “Occasionally, a filipodium and cytoneme”

Line 426 change “filipodia and cytonemes formation” to “filipodium and cytoneme formation”

Line 470 capitalize “representative”

Line 528 capitalize “representative"

Line 565 change “cell possesses” to “the cells possess”

Line 565-566 check grammar of “called as cell surface excess”

Line 609 change “from both flagellar base” to “from both the flagellar base”

Line 624 possibly change to “are indicated by asterisks (insets), and the letter (F) denotes filipodia.”

Line 655 change to “that are important players in parasite-parasite”

Line 672 change to “importance of the selected drying methods”

Line 741 maybe change to “with grants from”

Line 743 maybe change to “are PhD fellows”

**Do you want your identity to be public for this peer review?** For information about this choice, including consent withdrawal, please see our Privacy Policy

Reviewer #3: No

---

## [Editor Report · Acceptance letter]

PONE-D-25-35035R1

PLOS ONE

Dear Dr. Pereira-Neves,

I'm pleased to inform you that your manuscript has been deemed suitable for publication in PLOS ONE. Congratulations! Your manuscript is now being handed over to our production team.

Kind regards,

on behalf of

Dr. António Machado

Academic Editor

PLOS ONE